# Association between Activity and Brain-Derived Neurotrophic Factor in Patients with Non-Alcoholic Fatty Liver Disease: A Data-Mining Analysis

**DOI:** 10.3390/life11080799

**Published:** 2021-08-07

**Authors:** Ryuki Hashida, Dan Nakano, Sakura Yamamura, Takumi Kawaguchi, Tsubasa Tsutsumi, Hiroo Matsuse, Hirokazu Takahashi, Lynn Gerber, Zobair M. Younossi, Takuji Torimura

**Affiliations:** 1Division of Rehabilitation, Kurume University Hospital, Kurume 830-0011, Japan; hashida_ryuuki@med.kurume-u.ac.jp (R.H.); matsuse_hiroh@kurume-u.ac.jp (H.M.); 2Department of Orthopedics, Kurume University School of Medicine, Kurume 830-0011, Japan; 3Division of Gastroenterology, Department of Medicine, Kurume University School of Medicine, Kurume 830-0011, Japan; nakano_dan@med.kurume-u.ac.jp (D.N.); yamamura_sakura@med.kurume-u.ac.jp (S.Y.); tsutsumi_tsubasa@med.kurume-u.ac.jp (T.T.); tori@med.kurume-u.ac.jp (T.T.); 4Division of Metabolism and Endocrinology, Faculty of Medicine, Saga University, Saga 840-8502, Japan; takahas2@cc.saga-u.ac.jp; 5Center for Liver Disease, Department of Medicine, Inova Fairfax Hospital, Falls Church, VA 22042, USA; ngerber1@gmu.edu (L.G.); zobair.younossi@cldq.org (Z.M.Y.)

**Keywords:** steatosis, physical activity, patient-reported outcome measure, brain-derived neurotrophic factor, diabetes mellitus

## Abstract

Reduction in activity links to the development and progression of non-alcoholic fatty liver disease (NAFLD). Brain-derived neurotrophic factor (BDNF) is known to regulate an activity. We aimed to investigate the association between reduction in activity and BDNF in patients with NAFLD using data-mining analysis. We enrolled 48 NAFLD patients. Patients were classified into reduced (*n* = 21) or normal activity groups (*n* = 27) based on the activity score of the Chronic Liver Disease Questionnaire-NAFLD/non-alcoholic steatohepatitis. Circulating BDNF levels were measured using an enzyme-linked immunoassay. Factors associated with reduced activity were analyzed using decision-tree and random forest analyses. A reduction in activity was seen in 43.8% of patients. Hemoglobin A1c and BDNF were identified as negative independent factors for reduced activity (hemoglobin A1c, OR 0.012, *p* = 0.012; BDNF, OR 0.041, *p* = 0.039). Decision-tree analysis showed that “BDNF levels ≥ 19.1 ng/mL” was the most important classifier for reduced activity. In random forest analysis, serum BDNF level was the highest-ranked variable for distinguishing between the reduced and normal activity groups (158 valuable importance). Reduced activity was commonly seen in patients with NAFLD. Data-mining analyses revealed that BNDF was the most important independent factor corresponding with the reduction in activity. BDNF may be an important target for the prevention and treatment of NAFLD.

## 1. Introduction

Non-alcoholic fatty liver disease (NAFLD) is a major health problem worldwide [1]. NAFLD is a risk factor for liver cirrhosis, liver cancer, extrahepatic malignancies, and cardiovascular diseases [1]. A sedentary lifestyle is a well-known risk factor for NAFLD [2], and lifestyle modification, including maintaining physical activity, is the first-line therapy for NAFLD [3]. In fact, exercise improves insulin sensitivity, lipid metabolism, hepatic steatosis, and hepatic fibrosis in patients with NAFLD [4,5]. Moreover, maintaining physical activity is related to increased survival rates in patients with NAFLD [6,7]. Accordingly, activity assessment is important for the prevention and management of NAFLD.

Various methods are used to assess activity. The recently developed Chronic Liver Disease Questionnaire (CLDQ)-NAFLD/non-alcoholic steatohepatitis (NASH) is a useful, disease-specific, patient-reported outcome measure for patients with NAFLD [8,9]. In this context, the CLDQ-NAFLD/NASH has been sufficiently validated and widely used in international clinical trials [10,11]. The CLDQ-NAFLD/NASH includes 36 questions grouped into 6 domains of health-related quality of life, such as fatigue, systemic symptoms, emotional health, worry, abdominal symptoms, and activity. The lower score of the activity domain is associated with obesity, and the activity domain is able to capture physical function in patients with NAFLD [11].

Physical function is regulated by various factors. Metabolic disorders, including diabetes mellitus, are well-known factors associated with reduced physical function [12]. A meta-analysis showed that diabetes mellitus is associated with the risk of disabilities related to mobility and activities of daily living [13]. Sarcopenia is also a risk factor for reduced physical function and is induced by the disturbance of various inflammatory and anti-inflammatory cytokines, including interleukin (IL)-6, IL-1, IL-10, and IL-15 [14,15].

Moreover, mental health is known to regulate physical function [16]. A decrease in the levels of brain-derived neurotrophic factor (BDNF), a neurotrophic protein, is linked to neuronal atrophy, leading to depressive symptoms [17]. On the other hand, an increase in the BDNF level promotes neuronal differentiation and neuronal transmission [17]. BDNF is released from the brain during exercise [18]. BDNF is a myokine that increases in the muscles in response to exercise, and it promotes fatty acid oxidation [19,20]. BDNF is thought to play a role in physical performance via this mechanism. The circulating BDNF level is positively associated with the intensity of physical activity in patients who are overweight/obese [21]. However, the association between activity and BDNF levels remains unclear in patients with NAFLD.

Physical function is regulated by diverse factors. A data-mining analysis is an artificial intelligence approach to reveal factors, even if no a priori hypothesis has been imposed [22]. This approach allows us to discover hidden factors that cannot be identified by logistic regression analysis [23]. Decision tree analysis is a data-mining technique and identifies factors with priorities. Thus, decision tree analysis is useful to discover the most impacting factor associated with physical function. Random forest analysis is also a data-mining technique used to identify factors that distinguish between case and control groups by random sampling. A feature of random forest analysis is a high level of predictive accuracy with an estimation of the relative importance for each factor [24]. Recently, these data-mining techniques have been used to identify individuals at an increased risk of developing NAFLD [25] and prognostic algorithms for patients with NAFLD-related hepatocellular carcinoma [26]. However, these unique statistical techniques have never been used to investigate the factor associated with physical function.

Therefore, the aim of this study was to investigate the association between physical activity and BDNF levels in patients with NAFLD using data-mining analysis.

## 2. Results

### 2.1. Prevalence of Reduction in Activity in Patients with NAFLD

Among the six domains of CLDQ-NAFLD/NASH, fatigue was the most impacted domain. The activity was also an impacted domain. A reduction in activity was seen in 43.8% of patients with NAFLD (Figure 1).

### 2.2. Patient Characteristics

Patient characteristics are summarized in Table 1. There was no significant difference in age or BMI between the reduced and normal activity groups. Although the prevalence of women was significantly higher in the reduced activity group than in the normal activity group, no significant difference was seen in the liver frailty index, the prevalence of type 2 diabetes mellitus, or liver stiffness between the two groups (Table 1).

There was no significant difference in hemoglobin level, hepatic and renal function tests, or hemoglobin A1c (HbA1c) levels between the two groups. No significant difference was observed in serum levels of C-reactive protein, BDNF, myostatin, or inflammatory cytokines, such as IL-1, IL-6, IL-10, and IL-15, between the two groups (Table 1).

### 2.3. Independent Factors Associated with the Reduction in Activity in Patients with NAFLD

In a stepwise procedure, the following five factors were selected as explanatory variables: HbA1c, BDNF, liver frailty index, BMI, and sex. In a logistic regression analysis, HbA1c and BDNF levels were identified as independent negative factors associated with the reduction in activity in patients with NAFLD (Figure 2).

### 2.4. Decision-Tree Analysis for the Reduction in Activity

A decision-tree algorithm was created by using two variables to classify three groups of subjects for the reduction in activity (Figure 3). BDNF was selected as the most important classifier with an optimal cut-off of 19.1 ng/mL. In patients with BDNF levels < 19.1 ng/mL, all patients showed normal activity. On the other hand, in patients with BDNF levels ≥ 19.1 ng/mL, 53.6% of patients showed a reduction in activity. In such patients, HbA1c was selected as the second classifier with an optimal cut-off of 7.1%. Reduction in activity was seen in all patients with BDNF levels ≥ 19.1 ng/mL and HbA1c > 7.1%.

### 2.5. Random Forest Analysis for Distinguishing between Reduced and Normal Activity Groups

The results of the random forest analysis are summarized in rank order in Figure 4. The serum BDNF level was the highest-ranked variable for distinguishing between the reduced and normal activity groups. This was followed by HbA1c, IL-6, and liver stiffness (Figure 4).

## 3. Discussion

In this study, we demonstrated that reduction in the activity domain of CLDQ-NAFLD/NASH was seen in 43.8% of patients with NAFLD. In addition, we found that BDNF and HbA1c were independent negative factors for the reduction in activity. The data-mining analysis also revealed that the BNDF was the most important classifier for the reduction in activity. A higher serum level of BDNF was associated with a reduction in activity in patients with NAFLD.

The male-to-female ratio was significantly higher in the reduced activity group than in the normal activity group in this study. Sex was not an independent factor for reduced activity; this may be related to the low statistical power. However, Cohen et al. recently investigated gender differences in physical activity of general subjects and demonstrated that physical activity level is significantly higher in men than in women [27]. Moreover, in decision-tree analysis, sex was not identified as a classifier with the reduced activity. Furthermore, in random forest analysis, sex was the fifth-ranked variable for distinguishing between the reduced and normal activity groups in patients with NAFLD. Thus, in patients with NAFLD, the impact of sex on physical activity may be different from the general population and may have less impact on physical activity level.

In this study, we investigated factors associated with the reduced activity using various factors, including age, sex, BMI, HbA1c, liver stiffness, and inflammatory and anti-inflammatory cytokines, including IL-6, IL-1, IL-10, and IL-15. However, none of these inflammatory cytokines was identified as an independent factor for reduced activity. Taken together, our findings suggest that activity may be mainly regulated by factors other than hepatic fibrosis and inflammatory cytokines in patients with NAFLD.

We employed exploratory analyses to eliminate possible bias and found that BDNF and HbA1c were independent factors associated with the reduction in activity among patients with NAFLD. The data-mining analysis further revealed that HbA1c was the second classifier for reduced activity in this study. Although there was no significant difference in HbA1c value and BDNF levels at the baseline between the normal and reduced activity groups, the difference in the results between univariate and multivariate analyses can be explained by confounding factors [28]. Serum BDNF levels are known to be modulated by various factors, including age [29] and sex [30], and the presence of hypertension [31]. A previous study has also reported that type 2 diabetes mellitus is associated with unhealthy behaviors, including reduced physical activity and increased sedentary behaviors [32]. In patients with NAFLD, type 2 diabetes mellitus has been reported to be associated with a reduction in activity, which was evaluated by the CLDQ-NAFLD/NASH [8]. Thus, our findings were in good agreement with previous reports. Although the causal relationship between HbA1c and reduction in activity remains unclear, the following are possible explanations. Physical inactivity may cause a reduction in energy consumption, leading to a reduction in glucose metabolism [33]. Alternatively, an association between HbA1c and reduction in activity may be affected by confounding factors, such as sarcopenia and depression [34,35].

BDNF is reported to be a regulator of physical activity [19]. Previous animal studies have demonstrated a positive association between brain BDNF levels and physical activity in rats [36,37]. These findings led us to hypothesize a positive association between serum BDNF levels and activity in patients with NAFLD. However, serum BDNF levels were negatively associated with the activity level of patients with NAFLD in our study. We also investigated the correlation of serum BNDF levels and FIB-4 index, M2BPGi levels, and liver stiffness values. However, no strong correlation was seen between serum BNDF levels and these hepatic fibrosis markers (data not shown). Chan et al. previously reported that the serum BDNF level is higher in healthy subjects with sedentary behaviors, such as watching television [38]. Huang et al. reported that higher serum BDNF levels were associated with lower physical activity levels in adolescents [39]. Moreover, Suwa et al. reported that serum BDNF levels were high in patients who are obese and have type 2 diabetes mellitus [40]. Levinger et al. also reported that serum BDNF levels were positively correlated with serum triglyceride levels, insulin resistance, and the number of metabolic risk factors [41]. Furthermore, de Avila et al. reported that, in patients with hepatitis C virus infection, BDNF was negatively correlated with SF-36 physical functioning [42], which is significantly associated with the activity domain of the CLDQ-NAFLD/NASH [8].

We could not examine the causal relationship between BNDF level and reduction in activity in patients with NAFLD in this study. Although the biological significance of BNDF in the reduction in activity remains unclear, BDNF has been reported to be modulated by various factors, including race [43], depressive symptoms [44], cognitive function [45], and genetic polymorphism [46,47]. These factors might be involved in the negative association between serum BDNF levels and activity in this study. In addition, BDNF has been reported to directly affect various metabolic pathways. BDNF reduces appetite through the regulation of serotonergic neurons in animal models [48,49]. BDNF also improves insulin resistance and lipid metabolism through the upregulation of energy metabolism in a mouse model of diabetes mellitus [50]. Taken together, BDNF may have been upregulated in this study as a compensatory mechanism to improve metabolic dysfunction and insulin resistance in patients with NAFLD who have a reduction in activity.

We employed a data mining analysis, an artificial intelligence approach, and revealed that “BDNF level ≥ 19.1 ng/mL and HbA1c ≥ 7.1%” is a profile for the high prevalence of reduced activity in patients with NAFLD. Since a reduction in activity is a risk factor for disease progression and mortality of patients with NAFLD [6,7], patients with such a profile may be an important target for the prevention and treatment of NAFLD.

This study has several limitations. First, this was a single-center study with small sample size. Second, this was a nested case-control study lacking a healthy subjects group and time-course changes in BDNF level. Third, we stratified subjects into two groups. Fourth, activity was evaluated by the CLDQ-NAFLD/NASH alone and not by other questionnaires or activity monitoring, including the International Physical Activity Questionnaire or an accelerometer. Thus, further multicenter prospective studies with healthy control, multiple questionnaires, and activity monitoring are required to elucidate the association between BDNF and reduction in activity in patients with NAFLD.

In this study, we showed that a reduction in activity was seen in 43.8% of patients with NAFLD. Furthermore, data-mining analyses revealed that BDNF was the most important negative factor associated with the reduction in activity in patients with NAFLD. Since a reduction in activity is involved in the development and progression of NAFLD, BDNF may be an important target for the prevention and treatment of NAFLD.

## 4. Materials and Methods

### 4.1. Study Design and Ethics

We conducted a single-center, nested case-control study in Japan. The protocol conformed to the ethical guidelines of the 1975 Declaration of Helsinki, as reflected by prior approval from the institutional review board of the Kurume University School of Medicine (approval number 20226). This research was performed under the relevant guidelines and regulations. An opt-out approach was used to obtain informed consent from patients, and personal information was protected during data collection.

### 4.2. Subjects

We enrolled 48 consecutive outpatients with NAFLD who visited Kurume University Hospital between May 2019 and April 2020. The inclusion criteria were as follows: (1) age > 18 years, (2) answered the CLDQ-NAFLD/NASH, and (3) underwent liver stiffness measurement. Exclusion criteria were as follows: (1) complications of other major diseases that might bias their activity, including congestive heart failure, renal failure, respiratory disease, cancer, and orthopedic disease, and (2) any condition that could interfere with answering survey queries, including psychiatric or emotional problems, language, or cognitive difficulties.

### 4.3. Diagnosis of NAFLD

The diagnosis of NAFLD was established according to the Clinical Practice Guidelines for NAFLD/NASH [3,51,52]: (1) hepatic steatosis evaluated by liver biopsy, ultrasonography, computed tomography, or magnetic resonance imaging; (2) ethanol intake <20 g/day in women or <30 g/day in men; and (3) exclusion of other liver diseases, including hepatitis B virus, hepatitis C virus, autoimmune hepatitis, drug-induced liver disease, primary biliary cholangitis, primary sclerosing cholangitis, biliary obstruction, Wilson’s disease, and hemochromatosis.

### 4.4. Data Collection

All data were collected prospectively at the patients’ regular visits. The following information was obtained using a self-reported questionnaire: age, sex, and amount of daily alcohol intake. In the clinical review, we obtained the following data: body mass index (BMI), waist circumference, presence/absence of type 2 diabetes mellitus, hypertension, and dyslipidemia; these were diagnosed according to standard criteria [53]. We also assessed physical frailty using the liver frailty index, as previously reported [54].

### 4.5. Disease-Specific Patient-Reported Outcome Measure for Patients with NAFLD

The CLDQ-NAFLD/NASH is a disease-specific questionnaire that consists of 36 items grouped into 6 domains: fatigue, systemic symptoms, emotional health, worry, abdominal symptoms, and activity [8,10,11]. All participants were asked to complete the Japanese version of the CLDQ-NAFLD, which has previously been validated in Japan [9]. The questions were closed-ended, and the answers were provided using a Likert scale ranging from 1 (all the time) to 7 (none of the time), representing the frequency of clinical symptoms and emotional problems. The average score was calculated for each domain, and higher scores indicated a minimal frequency of symptoms and, consequently, a better health-related quality of life.

### 4.6. Assessment of Activity and Definition for Reduction in Activity

Activity was assessed by the activity domain of the CLDQ-NAFLD/NASH. Patients with activity domain scores < 6 were classified into the reduced activity group. Patients with activity domain scores ≥ 6 were classified into the normal activity group.

### 4.7. Biochemical Examination, Serum Levels of BDNF, and Serum Levels of Inflammatory Cytokines and Myokine

Blood samples were collected after overnight fasting. The following biochemical tests were performed: complete blood cell count, aspartate aminotransferase, alanine aminotransferase, alkaline phosphatase, γ-glutamyl transpeptidase, albumin, total bilirubin, total cholesterol, low-density lipoprotein cholesterol, triglycerides, fasting glucose, HbA1c, blood urea nitrogen, creatinine, estimated glomerular filtration rate, C-reactive protein, Mac-2 binding protein glycosylation isomer, alpha-fetoprotein, and des-gamma-carboxy prothrombin.

Serum levels of BDNF were measured using a Human BDNF enzyme-linked immunoassay (ELISA) kit according to the manufacturer’s instructions (R&D Systems, Inc., Minneapolis, MN, USA).

Serum levels of IL-1, IL-6, IL-10, IL-15, and myostatin were measured using a Human IL-1 ELISA Kit, a Human IL-6 ELISA Kit, a Human IL-10 ELISA Kit, a Human IL-15 ELISA Kit, or a Myostatin Quantikine ELISA Kit from R&D Systems, Inc. (Minneapolis, MN, USA), according to the manufacturer’s instructions.

### 4.8. Measurement for Liver Stiffness and Controlled and Hepatic Steatosis

Liver stiffness and hepatic steatosis were evaluated under fasting conditions for more than 4 h by vibration-controlled transient elastography and controlled attenuation parameters using FibroScan^®^ (Echosens, Paris, France), respectively. The median value of 10 valid measurements was calculated, and the results of elasticity and controlled attenuation parameters were expressed in kiloPascals (kPa) and dB/m, respectively. A valid result included at least 10 valid measurements with an interquartile range (IQR)/median value of <30%, as previously described [55].

### 4.9. Statistical Analysis

Data are expressed as median (interquartile range), range, or number. The differences between the reduction and no reduction in activity groups were analyzed using Wilcoxon rank-sum tests. Factors associated with the reduction in activity were analyzed using logistic regression analysis and data mining techniques. The statistical methods are described in detail below.

Independent factors associated with reduced activity levels were analyzed using logistic regression analysis, as previously described. Explanatory variables were selected in a stepwise manner, minimizing the Bayesian information criterion, as previously described [26]. Data were expressed as odds ratios and 95% confidence intervals.

A decision-tree algorithm was constructed to reveal profiles associated with reduced activity, as previously described [26]. The initial classifier was the most important factor associated with reduced activity. Subjects were classified according to the cut-off values indicated for each variable. The cut-off value was determined by the decision-tree analysis, which gives the best split of valuables [56,57].

A random forest analysis was used to identify factors distinguishing between the reduced and normal activity groups on an ordinal scale, as previously described [23]. The variable importance value, which reflects the relative contribution of each variable to the model, was estimated by randomly permuting its values and recalculating the predictive accuracy of the model.

All statistical analyses were performed using JMP Pro^®^ 15 (SAS Institute Inc., Cary, NC, USA). All *p* values were 2-tailed, and a value < 0.05 was considered statistically significant.

## Figures and Tables

**Figure 1 life-11-00799-f001:**
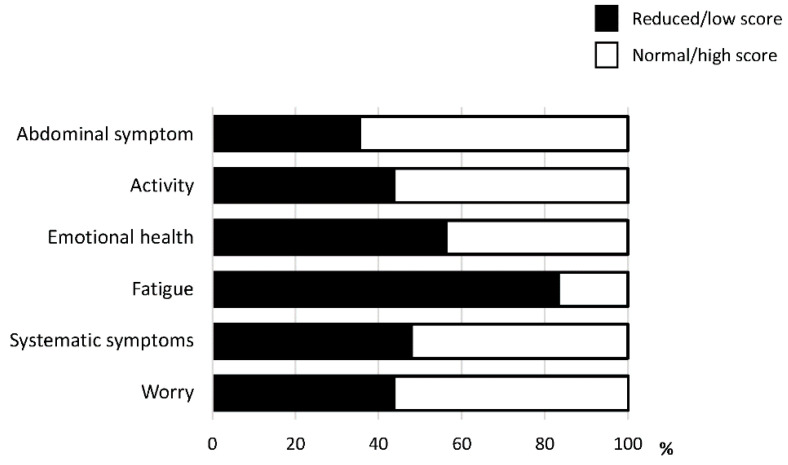
The prevalence of reduction in each domain of the Chronic Liver Disease Questionnaire-non-alcoholic fatty liver disease/non-alcoholic steatohepatitis (CLDQ-NAFLD/NASH).

**Figure 2 life-11-00799-f002:**
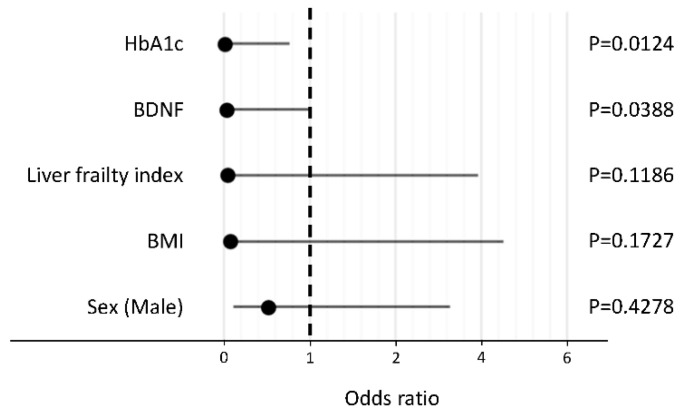
Independent factors associated with the reduction in activity in patients with non-alcoholic fatty liver disease. Abbreviations: HbA1c, hemoglobin A1c; BDNF, brain-derived neurotrophic factor; BMI, body mass index.

**Figure 3 life-11-00799-f003:**
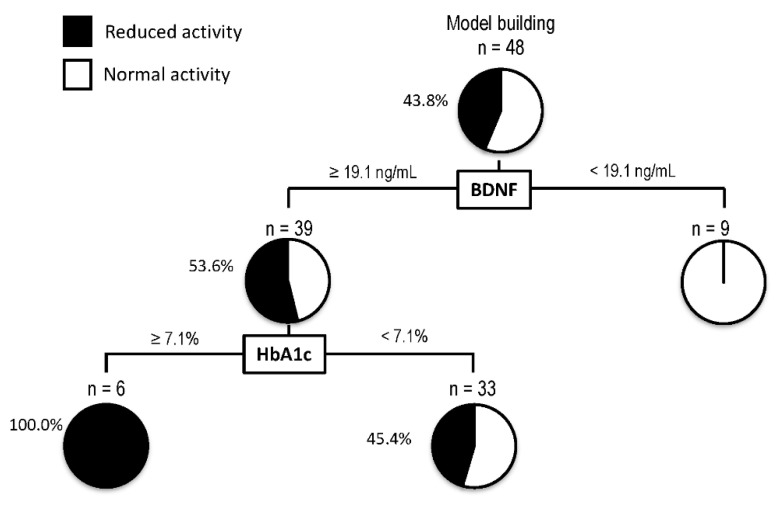
Decision-tree analysis for the reduction in activity in patients with non-alcoholic fatty liver disease. The subjects were classified according to the indicated cut-off values of the variables. The pie graphs indicate the proportion of patients with reduced activity (black) and normal activity (white). Abbreviations: HbA1c, hemoglobin A1c; BDNF, brain-derived neurotrophic factor.

**Figure 4 life-11-00799-f004:**
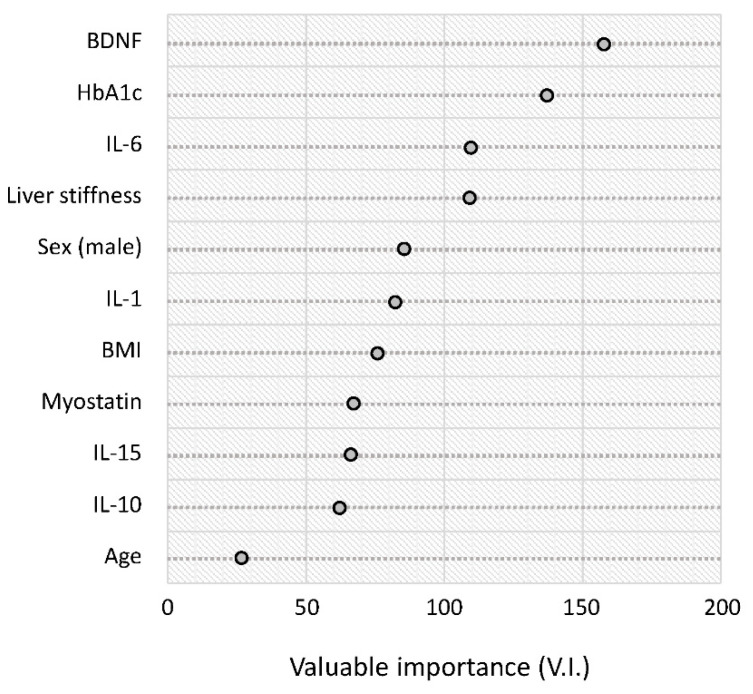
Random forest analysis for distinguishing between the reduced and normal activity groups. Variable importance is a general measure of the contribution of each variable in distinguishing the classes. Abbreviations: HbA1c, hemoglobin A1c; BDNF, brain-derived neurotrophic factor; IL, interleukin; BMI, body mass index.

**Table 1 life-11-00799-t001:** Comparison of patients’ characteristics between the patients with no impairment and impairment groups.

	Normal Activity Group		Reduced Activity Group		
	Median (IQR)	Range (Min-Max)	Median (IQR)	Range (Min-Max)	*p*
Number	27	N/A	21	N/A	N/A
Age (years)	58 (45–65)	22–82	59 (38–68)	28–76	0.9917
Sex (female/male)	51.9%/48.1% (14/13)	N/A	23.8%/76.2% (5/16)	N/A	0.0487
Body mass index (kg/m^2^)	24.8 (22.4–29.4)	18.0–39.7	27.2 (24.9–30.0)	21.0–47.3	0.1241
Waist circumference (Large/Normal)	63.0%/37.0% (17/10)	N/A	80.9%/19.1% (17/4)	N/A	0.1737
Liver Frailty Index (Pre-frail/Robust)	19.2%/80.8% (5/21)	N/A	35.0%/65.0% (7/13)	N/A	0.2273
Type 2 Diabetes mellitus (Presence/Absence)	22.2%/77.8% (6/21)	N/A	38.1%/61.9% (8/13)	N/A	0.2300
Hypertension (Presence/Absence)	77.8%/22.2% (21/6)	N/A	57.1%/42.9% (12/9)	N/A	0.1260
Dyslipidemia (Presence/Absence)	48.2%/51.8% (13/14)	N/A	52.4%/47.6% (11/10)	N/A	0.7711
Hepatic steatosis (db/m)	308 (271–325)	236–400	297 (267–354)	191–400	0.7316
Liver stiffness (kPa)	5.3 (4.2–8.8)	2.0–51.4	7.9 (4.4–11.8)	3.2–40.3	0.2162
Liver stiffness (Normal/High)	59.3%/40.7% (16/11)	N/A	47.6%/52.4% (10/11)	N/A	0.4220
Biochemical examinations					
Red blood cell count (×106/µL)	4.77 (4.45–5.23)	4.00–5.76	4.69 (4.33–5.34)	3.90–6.11	0.7006
Hemoglobin (g/dL)	14.5 (13.6–16.1)	10.0–17.7	14.7 (13.6–16.1)	10.2–19.2	0.7550
White blood cell count (×103/µL)	5.4 (4.6–7.4)	2.3–9.3	6.1 (5.3–7.8)	3.8–11.3	0.2317
Platelet count (×104/µL)	21.9 (18.0–27.6)	10.1–41.5	22.1 (17.4–26.1)	11.4–30.2	0.8761
AST (U/L)	34 (24–47)	21–115	40 (28–54)	15–96	0.6103
ALT (U/L)	44 (22–80)	8–223	60 (27–95)	9–148	0.3603
FIB-4 index	1.57 (0.91–2.41)	0.35–4.07	1.23 (0.97–2.06)	0.34–4.71	0.7870
ALP (U/L)	230 (173–280)	92–496	252 (217–327)	149–423	0.1834
GGT (U/L)	53 (31–100)	7–244	82 (48–154)	10–538	0.2124
Albumin (g/dL)	4.5 (4.4–4.7)	3.2–5.2	4.4 (4.2–4.6)	3.3–5.0	0.1615
Total bilirubin (mg/dL)	0.9 (0.7–1.1)	0.4–1.7	0.9 (0.8–1.4)	0.4–1.8	0.3109
Total cholesterol (mg/dL)	189 (177–219)	102–308	199 (165–221)	110–287	0.7700
LDL cholesterol (mg/dL)	122 (110–137)	63–205	131 (105–155)	72–210	0.4039
Triglycerides (mg/dL)	119 (81–156)	52–241	108 (80–119)	44–338	0.3825
Fasting glucose (mg/dL)	104 (97–110)	89–142	110 (97–154)	91–196	0.1250
HbA1c (%)	6.0 (5.7–6.2)	5.3–7.0	6.2 (5.7–7.2)	5.5–8.5	0.1535
BUN (mg/dL)	13 (12–17)	8–22	15 (13–19)	9–33	0.1526
Creatinine (mg/dL)	0.69(0.54–0.76)	0.46–1.00	0.74(0.59–0.84)	0.43–7.66	0.2181
eGFR (mL/min/1.73 m^2^)	85.3 (71.9–96.5)	56.3–126.9	84.5 (69.1–108.1)	6.1–140.6	0.9659
CRP (mg/dL)	0.10 (0.04–0.18)	0.04–0.92	0.09 (0.05–0.28)	0.04–0.91	0.7793
M2BPGi (C.O.I.)	0.73 (0.56–1.63)	0.26–3.70	0.72 (0.55–1.08)	0.36–8.02	0.8357
AFP (ng/mL)	3.3 (2.2–4.4)	1.2–12.7	3.5 (2.2–5.0)	1.5–7.8	0.8426
DCP (mAU/mL)	22 (18–27)	15–36	23 (17–28)	13–38	0.8644
BDNF (ng/mL)	22.2 (17.4–28.4)	10.0–37.6	25.0 (20.4–29.1)	19.1–37.6	0.2240
Cytokines					
Myostatin (ng/mL)	3.03 (1.90–3.45)	1.25–7.07	3.31 (2.31–4.79)	1.33–7.81	0.3211
IL-10 (pg/mL)	0.52 (0.14–1.06)	0–6.17	0.79 (0.13–1.43)	0–4.36	0.4543
IL-15 (pg/mL)	2.27 (2.13–2.53)	1.23–3.79	2.35 (2.05–2.65)	1.16–3.37	0.6934
IL-6 (pg/mL)	5.33 (1.64–35.4)	0.92–301.2	9.76 (1.99–129.9)	1.23–212.3	0.1458
IL-1 (pg/mL)	399.1 (310.7–803.6)	188.6–2011.5	628.4 (377.7–1094.8)	199.6–1918.9	0.1117

Note. Data are expressed as median (interquartile range [IQR]), range, or number. Abbreviations: N/A, not applicable; AST, aspartate aminotransferase; ALT, alanine aminotransferase; FIB-4, fibrosis-4; ALP, alkaline phosphatase; GGT, gamma-glutamyl transpeptidase; LDL cholesterol, low-density lipoprotein cholesterol; HbA1c, hemoglobin A1c; BUN, blood urea nitrogen; eGFR, estimated glomerular filtration rate; CRP, C-reactive protein; M2BPGi, Mac-2 binding protein glycosylation isomer; AFP, alpha-fetoprotein; DCP, des-gamma-carboxy prothrombin; BDNF, Brain-derived neurotrophic factor; IL, interleukin.

## Data Availability

Data are available upon reasonable request.

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
