# Peer review of "Association between Activity and Brain-Derived Neurotrophic Factor in Patients with Non-Alcoholic Fatty Liver Disease: A Data-Mining Analysis"

_life, 2021, doi:10.3390/life11080799_

Round 1

Reviewer 1 Report

The manuscript by Hashida et al., tried to link/associate reduced activity of the NAFLD patients with the higher circulating levels of brain derived factor (BDNF). Making use of data mining analysis, authors tried to establish that BDNF is the one of the critical molecules associated with the3 reduction of activity and therefore it could be targeted for therapeutic purpose.

Major concerns:

  1. Why no control (Healthy volunteer group) was not included?
  2. Although authors acclaimed that BDNF levels are higher in patients with reduced activity, the data in Table 1 shows that BDNF level was 22 in normal activity group compared to 25 in reduced activity group. It is hard to believe that this much increase with p value =0.2240 is critical for reduced activity in NAFLD.
  3. Although IL1 (399 vs 628.4) and IL-6 (5.33 vs 9.76) levels were higher in reduced activity group, why authors concentrated only on BDNF is not clear.
  4. NAFLD is a multi-hit process, therefore a follow up study by measuring circulating BDNF at various time points of one particular patient could be more informative.

Author Response

To REVIEWER 1

Thank you very much for your letter regarding our manuscript (life-1107884). We appreciate your comments, which have helped us to improve our manuscript. In line with your comments, please find below our response.

Comment 1: Why no control (Healthy volunteer group) was not included?

Answer: As you pointed out, by the addition of a control group (healthy volunteer group), an association between circulating BDNF level and activity may be further supported. However, this study is designed as a nested case-control study of the NAFLD cohort. It is difficult to enroll healthy volunteers in this study at present. Therefore, we added this issue as a limitation of this study (line 203).

Comment 2: Although authors acclaimed that BDNF levels are higher in patients with reduced activity, the data in Table 1 shows that BDNF level was 22 in normal activity group compared to 25 in reduced activity group. It is hard to believe that this much increase with p value =0.2240 is critical for reduced activity in NAFLD.

Answer: We appreciate your comment. As you pointed out, there was no significant difference in serum BDNF level between the normal activity and reduced activity group in univariate analysis (Wilcoxon rank-sum test; Table 1). However, in multivariate analysis (logistic regression analysis), a higher BDNF level was an independent factor associated with the reduced activity (Figure 2 and Figure 3). The difference in results between the Wilcoxon rank-sum test and logistic regression analysis can be explained by confounding factors [1]. Serum BDNF levels are known to be modulated by various factors including age [2], sex [3], and presence of hypertension [4]. Wilcoxon rank-sum test evaluates the difference in the objective variable between two groups with no consideration of confounding factors. However, logistic regression analysis identifies an independent factor associated with the objective variable with consideration of confounding factors. We have added the above descriptions in the Discussion section (line 166-170). Again, we appreciate that your comment, which has helped us to improve our manuscript.

Comment 3: Although IL1 (399 vs 628.4) and IL-6 (5.33 vs 9.76) levels were higher in reduced activity group, why authors concentrated only on BDNF is not clear.

Answer: We appreciate your comment. As you indicated serum IL-1 and IL-6 levels were higher in the reduced activity group than those in the normal activity group (Table 1). However, these factors were not identified as independent factors associated with the reduced activity in logistic regression analysis and decision-tree analysis. On the other hand, the serum BDNF level was identified as an independent factor associated with the reduced activity (Figures 2 and 3). In addition, the BDNF level was the highest-ranked variable for distinguishing between the reduced and normal activity groups (Figure 4). Thus, based on the results of logistic regression analysis and data-mining analyses, we focused on the BDNF level in this study.

Comment 4: NAFLD is a multi-hit process, therefore a follow up study by measuring circulating BDNF at various time points of one particular patient could be more informative.

Answer: We agree with your comment. It is important to evaluate time-course changes in serum BDNF level. Unfortunately, the number of patients is limited for assessment of the time-course changes in serum BDNF level. Thus, we have added this issue as a limitation of this study. (line 204).

References

1    Skelly, A.C.; Dettori, J.R., Brodt, E.D. Assessing bias: the importance of considering confounding. Evid Based Spine Care J. 2012; 3: 9-12.

2    Erickson, K.I.; Prakash, R.S.; Voss, M.W.; Chaddock, L.; Heo, S.; McLaren, M.; Pence, B.D.; Martin, S.A.; Vieira, V.J.; Woods, J.A.; et al. Brain-derived neurotrophic factor is associated with age-related decline in hippocampal volume. J Neurosci. 2010; 30: 5368-75.

3    Chan, C.B., Ye, K. Sex differences in brain-derived neurotrophic factor signaling and functions. J Neurosci Res. 2017; 95: 328-35.

4    Amoureux, S.; Lorgis, L.; Sicard, P.; Girard, C.; Rochette, L., Vergely, C. Vascular BDNF expression and oxidative stress during aging and the development of chronic hypertension. Fundam Clin Pharmacol. 2012; 26: 227-34.

Reviewer 2 Report

In this single-center study, the Authors analyzed the serum level of potential markers/explanatory factors for the reduced activity in 48 subject with NAFLD, with special focus on BDNF. An inverse correlation between BDNF and HbA1c blood levels and reduced physical activity was suggested. 19.1 ng/mL BDNF and 7.1% HbHA1c have been found as cut-off limits.

As reported by the Authors, some limitation to the work exists.

As first, the number of subject enrolled, that is only 48, looks to be really limiting. As actually stratified (only in “normal activity” and “reduced activity”), the two groups look to much superimposed, and lacking important controls. This will introduce potential biases in the analysis, finally leading to a reduced capability of extrapolating the most correct factors responsible for the reduced activity in subjects with NAFLD. It is required to increase the number of subjects enrolled to better stratify the subjects and gain the adequate statistical significance for improving this nice work.

Most importantly, a potential bias in the experimental plan exist. The 48 subject have been immediately divided based on the range of physical activity. If the hypotheses that BDNF serum level is associated with the level of physical activity is true (like the literature reports), automatically the analysis of the role of BDNF in this study is biased (pushed in this respect).

Indeed, both the group as they currently are, present a large variability in several parameters, what is especially clear observing the range data in Table 1. A further stratification of the subjects is required to improve the meaning of the work.  

  • As example, excessively large looks to be the male /female ratio in the “reduced activity” group. This avoid the possibility to understand if the increased fatigue is sex related (the unsuccessful result in Figure 2 may be related to the low statistical power).
  • Based on the BMI, even underweight subjects have been enrolled, up to severely obese peoples.
  • 1) Can the authors explain the potential causes of NAFLD in subject with a normal BMI?
  • 2) In the abstract and in the introduction the Authors reported that obesity and BDNF are linked. Thus, normal weight persons have to be divided from the overweight peoples, and then each group analyzed based on the activity score. Lean peoples deserve as control able to dissect NAFLD from obesity.
  • While it is clear the rational for the cut-off value for the decision-tree analysis concerning BDNF (table 1), the same is not so apparent concerning HbAc1. How the Authors decided for a cut-off of 7.1%?
  • Question: Total bilirubin in both groups reached the 1.7 – 1.8 mg/dL. This value is unusual. How can the authors explain the data?

Discussion:

Please modify the first sentence of the discussion “in this study, we demonstrated that reduction of activity was seen in 43,8% of patients with NAFLD”. This is not truly a result, but your approach for the classification of the subjects enrolled.

Not fully convincing is the explanation for the:

  • Link between liver fibrosis (lines 150-160), activity and BDNF. Still annoying is the limited number of patients enrolled and the over imposed group here analyzed, biasing or limiting the analysis. Please or perform an additional analysis comparing NAFLD without advanced fibrosis to NAFLD with advanced fibrosis, or remove the sentence.
  • Strange results about BDNF (higher BDNF is, lower is the physical activity). We agree with the sentence from line 174 to line 188. As you reported, large evidence supports the beneficial effects of BDNF increase on liver (steatosis, NASH), appetite, insulin resistance, lipid profile, etc. The suggested explanation of the controversial result in this work (lines 198-200), looks overstated, in absence of a direct evidence of an altered TrbK expression/variant in the subjects that you analyzed. Alternative explanations may reside in the population (Chinese vs European; adult vs pediatric, as example). Thus, please, enlarge the section of the discussion by adding clinical publications reporting the opposite results in respect to yours, and suggesting potential alternative explanations.

Author Response

To REVIEWER 2

Thank you very much for your letter regarding our manuscript (life-1107884). We appreciate your comments, which have helped us to improve our manuscript. In line with your comments, please find below our response.

Comment 1: As first, the number of subject enrolled, that is only 48, looks to be really limiting. As actually stratified (only in “normal activity” and “reduced activity”), the two groups look to much superimposed, and lacking important controls. This will introduce potential biases in the analysis, finally leading to a reduced capability of extrapolating the most correct factors responsible for the reduced activity in subjects with NAFLD. It is required to increase the number of subjects enrolled to better stratify the subjects and gain the adequate statistical significance for improving this nice work.

Answer: As you pointed out, we agree that a small sample size (n=48), stratification for two groups (normal activity or reduced activity), and lacking control group are potential biases in the analysis. However, this study is designed as a nested case-control study of the NAFLD cohort and we are not able to increase the number of enrolled subjects at present. It is also difficult to enroll healthy control in this study at present. Therefore, we added these issues as limitations of this study (lines 203-204).

Comment 2: Most importantly, a potential bias in the experimental plan exist. The 48 subject have been immediately divided based on the range of physical activity. If the hypotheses that BDNF serum level is associated with the level of physical activity is true (like the literature reports), automatically the analysis of the role of BDNF in this study is biased (pushed in this respect).

Answer: We appreciate your comment. As you pointed out, this study has a potential bias in the experimental plan. Therefore, we investigated factors associated with the reduced activity by exploratory multivariate analyses using various factors including age, sex, BMI, HbA1c, liver stiffness, and various cytokines. As a result, the association between reduced activity and serum BDNF level was confirmed by two types of exploratory analyses such as decision-tree and random forest analyses. Thus, we tried to eliminate possible bias by using various factors and multiple exploratory analyses. We have added the above descriptions in the revised manuscript (lines 161-163).

Comment 3: As example, excessively large looks to be the male/female ratio in the “reduced activity” group. This avoid the possibility to understand if the increased fatigue is sex related (the unsuccessful result in Figure 2 may be related to the low statistical power).

Answer: As you pointed out, the male-to-female ratio was significantly higher in the reduced activity group than in the normal activity group (P=0.0487). The unsuccessful result in Figure 2 may be related to the low statistical power as you suggested. However, Cohen et al. recently investigated gender difference in physical activity of general subjects and demonstrated that physical activity level is significantly higher in men than in women [1]. Moreover, in decision-tree analysis, sex was not identified as a classifier with the reduced activity in patients with NAFLD (Figure 3). Furthermore, in random forest analysis, sex (male) was the fifth-ranked variable for distinguishing between the reduced and normal activity groups in patients with NAFLD (Figure 4). Thus, in patients with NAFLD, the impact of sex on physical activity may be different from the general population and may have less impact on physical activity level. We have added the above discussion in the revised manuscript (lines 151-160).

Comment 4: Based on the BMI, even underweight subjects have been enrolled, up to severely obese peoples.

Answer: As you pointed out, we enrolled obese subjects as well as lean subjects in this study. Recently, a systematic review reported that, within the NAFLD population, 19.2% of people were lean and 40.8% were non-obese worldwide [2]. Moreover, Hiose et al. investigated a long-term outcome of NAFLD and reported that BMI was not associated with either mortality or liver-related events in patients with NAFLD [3]. Thus, lean/non-obese subjects were not a minor subtype in the NAFLD population and the prognosis of lean/non-obese NAFLD is comparable to that of obese NAFLD. Accordingly, we enrolled patients with NAFLD regardless of BMI. We would appreciate it if you could understand the inclusion of subjects regardless of BMI based on the above rationale.

Comment 5: Can the authors explain the potential causes of NAFLD in subject with a normal BMI?

Answer: We appreciate your comments. We have examined the difference in factors between the non-obese and obese NAFLD group (Please find the below Additional Table 1). There was no significant difference between the two groups in age and sex. Abdominal circumference was significantly lower in the non-obese NAFLD group than in the obese NAFLD group. However, no significant difference was seen in the prevalence of diabetes, hypertension, and hyperlipidemia between the two groups. Although causes of non-obese NAFLD remain unclear in this study, sarcopenia and alterations in gut microbiota have been reported to be pathophysiological factors associated with the development of non-obese NAFLD [4, 5]. In addition, the transmembrane 6 superfamily member 2 (TM6SF2) gene polymorphism has been reported to be associated with non-obese NAFLD [6]. Therefore, these factors may be potential causes of NAFLD in subjects with a normal BMI.

Additional Table 1. Comparison of patients’ characteristic between the patients with the non-obese and obese groups

Non-obese group

Obese group

Median (IQR)

Range

(min-max)

Median (IQR)

Range

(min-max)

P

Number

11

N/A

37

N/A

N/A

Age (years)

59 (45–69)

26–82

58 (43–66)

22–76

0.7128

Sex (female/male)

63.6%/36.4%

(7/4)

N/A

32.4%/67.6%

(12/25)

N/A

0.0632

Waist circumference (cm)

81.0 (79.0–83.5)

75–92

97.9 (90.5–105)

80–145

<0.0001

Liver Frailty Index

2.97 (2.63–3.1)

1.92–3.15

2.93 (2.66–3.32)

1.82–4.49

0.4479

Type 2 Diabetes mellitus (Presence/Absence)

9.1%/90.9%

(1/10)

N/A

35.1%/64.9%

(13/24)

N/A

0.0952

Hypertension (Presence/Absence)

45.5%/54.5%

(5/6)

N/A

75.7%/24.3%

(28/9)

N/A

0.0576

Dyslipidemia (Presence/Absence)

63.6%/36.4%

(7/4)

N/A

46.0%/54.0%

(17/20)

N/A

0.3029

Hepatic steatosis (db/m)

278 (260–308)

236–325

312 (275.5–353.5)

191–400

0.0299

Liver stiffness (kPa)

4.4 (3.8–4.7)

3.1–10.1

8 (4.7–11.2)

2–51.4

0.0087

Liver stiffness (Normal/High)

59.3%/40.7% (16/11)

N/A

47.6%/52.4% (10/11)

N/A

0.4220

Biochemical examinations

Red blood cell count (×106/µL)

4.68 (4.34–4.89)

4.0–5.69

4.93 (4.44–5.34)

3.9–6.11

0.1935

Hemoglobin (g/dL)

13.9 (13.2–14.6)

12.6–15.1

14.7 (13.7–16.5)

10.0–19.2

0.0347

Platelet count (×104/µL)

21.5 (19.4–30.2)

13.0–31.0

21.9 (16.7–25.6)

10.1–41.5

0.3973

AST (U/L)

27 (23–29)

21–47

41 (31–59.5)

15–115

0.0074

ALT (U/L)

24 (17–38)

8–88

52 (29.5–95)

9–223

0.0052

FIB-4 index

1.19 (0.93–2.06)

0.41–3.17

1.36 (0.94–2.22)

0.34–4.71

0.6411

ALP (U/L)

230 (197–347)

139–419

246 (173–290.5)

92–496

0.9609

GGT (U/L)

73 (20–119)

7–266

63 (43–115)

10–538

0.7313

Albumin (g/dL)

4.6 (4.3–4.7)

3.2–4.8

4.5 (4.2–4.7)

3.3–5.2

0.5959

Total bilirubin (mg/dL)

0.9 (0.7–1)

0.4–1.4

0.9 (0.7–1.3)

04–1.8

0.8533

Total cholesterol (mg/dL)

193.5 (179.8–223.3)

102–260

193 (169.2–219.5)

110–308

0.9529

LDL cholesterol (mg/dL)

1717 (106–137)

63–177

125 (110.8–147.3)

72–210

0.5718

Triglycerides (mg/dL)

102 (63.5–125)

51–137

1115 (84.5–162.8)

44–338

0.1346

Fasting glucose (mg/dL)

112 (102–121)

102–121

628 (91–1164)

91–1164

1.0000

HbA1c (%)

5.9 (5.7–6.1)

5.3–6.4

6.1 (5.7–6.5)

5.5–8.5

0.1973

BUN (mg/dL)

14 (9.8–17.5)

8–19

14 (12–17.5)

9–33

0.3402

Creatinine (mg/dL)

0.66 (0.52–0.81)

0.46–0.82

0.69 (0.59–0.8)

0.43–7.66

0.4584

eGFR (mL/min/1.73 m2)

75.7 (70.7–80.7)

70.7–80.7

58.7 (6.1–111.2)

6.1–111.2

1.0000

CRP (mg/dL)

0.04 (0.04–0.27)

0.04–0.89

0.12 (0.06–0.24)

0.04–0.92

0.0384

M2BPGi (C.O.I)

0.6 (0.3–0.97)

0.26–2.03

0.73 (0.56–1.19)

0.36–8.02

0.3957

AFP (ng/mL)

4.3 (2.8–4.4)

1.2–11.5

3.1 (2.1–4.7)

1.4–12.7

0.2908

DCP (mAU/mL)

21 (15.8–22.5)

15–28

23 (19–29)

13–38

0.0865

BDNF (ng/mL)

24.6 (18.7–26.1)

15.0–37.6

23.0 (19.6–30.1)

10.0–37.6

0.7498

Cytokines

Myostatin (ng/mL)

2.78 (1.61–3.21)

1.34–5.12

3.16 (2.48–4.65)

1.23–7.81

0.1125

IL-10 (pg/mL)

0.79 (0.35–1.85)

0–6.17

0.65 (0.12–1.16)

0–4.36

0.5167

IL-15 (pg/mL)

2.27 (1.71–2.8)

1.23–3.79

2.29 (2.07–2.58)

1.16–3.37

0.8459

IL-6 (pg/mL)

2.35 (1.19–49.67)

0.92–125.98

7.05 (1.71–66.38)

0.96–301.18

0.4370

IL-1 (pg/mL)

385.6 (280.6–472.6)

188.6–964.4

576.9 (321.9–957.2)

199.6–2011.5

0.1279

Comment 6: In the abstract and in the introduction the Authors reported that obesity and BDNF are linked. Thus, normal weight persons have to be divided from the overweight peoples, and then each group analyzed based on the activity score. Lean peoples deserve as control able to dissect NAFLD from obesity.

Answer: We apologize for the unclear statement. BDNF level is linked to the intensity of physical activity in patients with who are overweight/obese, but not obesity itself [7]. In the revised manuscript, we clearly stated this issue (lines 68-70).

Comment 7: While it is clear the rational for the cut-off value for the decision-tree analysis concerning BDNF (table 1), the same is not so apparent concerning HbAc1. How the Authors decided for a cut-off of 7.1%?

Answer: We apologize for the insufficient description of the cut-off value for BDNF and HbA1c. The cut-off value was determined by the decision-tree analysis. The decision-tree analysis is a data-mining analysis, which gives the best split of valuable [8, 9]. We have added this point in the revised manuscript (lines 309-310).

Comment 8: Question: Total bilirubin in both groups reached the 1.7 – 1.8 mg/dL. This value is unusual. How can the authors explain the data?

Answer: As you suggested, the highest levels of total bilirubin were unusual. Since no patient with decompensated liver cirrhosis was enrolled in this study, the hyperbilirubinemia may be due to constitutional jaundice. We appreciate your careful peer review.

Comment 9: Please modify the first sentence of the discussion “in this study, we demonstrated that reduction of activity was seen in 43,8% of patients with NAFLD”. This is not truly a result, but your approach for the classification of the subjects enrolled.

Answer: We appreciate your proper suggestion. As you pointed out, activity was assessed by the activity domain of CLDQ-NAFLD/NASH in this study. According to your suggestion, we have revised the sentence as follows: “in this study, reduction of the activity domain of CLDQ-NAFLD/NASH was seen in 43,8% of patients with NAFLD.” (lines 145-146).

Comment 10: Not fully convincing is the explanation for the: Link between liver fibrosis (lines 150-160), activity and BDNF. Still annoying is the limited number of patients enrolled and the over imposed group here analyzed, biasing or limiting the analysis. Please or perform an additional analysis comparing NAFLD without advanced fibrosis to NAFLD with advanced fibrosis, or remove the sentence.

Answer: We appreciate your comment. As you pointed out, we did not provide sufficient data for the explanation for the link between liver fibrosis, activity, and BDNF level. Since the number of patients with advanced fibrosis was limited, we have removed this paragraph in the revised manuscript according to your suggestion.

Comment 11: Not fully convincing is the explanation for the: Strange results about BDNF (higher BDNF is, lower is the physical activity). We agree with the sentence from line 174 to line 188. As you reported, large evidence supports the beneficial effects of BDNF increase on liver (steatosis, NASH), appetite, insulin resistance, lipid profile, etc. The suggested explanation of the controversial result in this work (lines 198-200), looks overstated, in absence of a direct evidence of an altered TrbK expression/variant in the subjects that you analyzed. Alternative explanations may reside in the population (Chinese vs European; adult vs pediatric, as example). Thus, please, enlarge the section of the discussion by adding clinical publications reporting the opposite results in respect to yours, and suggesting potential alternative explanations.

Answer: We appreciate your comment. As you suggested, our explanation of the controversial result is not based on direct evidence and is an overstatement. Following your suggestion, we have revised this paragraph by replacing alternative explanations as follows: “We could not examine the causal relationship between BNDF level and reduction of activity in patients with NAFLD in this study. Although the biological significance of BNDF in the reduction of activity remains unclear, BDNF has been reported to be modulated by various factors including race [10], depressive symptoms [11], cognitive function [12], and genetic polymorphism [13, 14]. These factors might be involved in the negative association between serum BDNF levels and activity in this study.” (lines 196-201). Again, we appreciate your constructive comments, which have helped us to improve our manuscript.

References

1    Cohen, D.A.; Williamson, S., Han, B. (2020) Gender Differences in Physical Activity Associated with Urban Neighborhood Parks: Findings from the National Study of Neighborhood Parks. Womens Health Issues.

2    Ye, Q.; Zou, B.; Yeo, Y.H.; Li, J.; Huang, D.Q.; Wu, Y.; Yang, H.; Liu, C.; Kam, L.Y.; Tan, X.X.E.; et al. (2020) Global prevalence, incidence, and outcomes of non-obese or lean non-alcoholic fatty liver disease: a systematic review and meta-analysis. Lancet Gastroenterol Hepatol 5: 739-52.

3    Hirose, S.; Matsumoto, K.; Tatemichi, M.; Tsuruya, K.; Anzai, K.; Arase, Y.; Shiraishi, K.; Suzuki, M.; Ieda, S., Kagawa, T. (2020) Nineteen-year prognosis in Japanese patients with biopsy-proven nonalcoholic fatty liver disease: Lean versus overweight patients. PLoS One 15: e0241770.

4    Shida, T.; Oshida, N.; Suzuki, H.; Okada, K.; Watahiki, T.; Oh, S.; Kim, T.; Isobe, T.; Okamoto, Y.; Ariizumi, S.I.; et al. (2020) Clinical and anthropometric characteristics of non-obese non-alcoholic fatty liver disease subjects in Japan. Hepatol Res 50: 1032-46.

5    Duarte, S.M.B.; Stefano, J.T.; Miele, L.; Ponziani, F.R.; Souza-Basqueira, M.; Okada, L.; de Barros Costa, F.G.; Toda, K.; Mazo, D.F.C.; Sabino, E.C.; et al. (2018) Gut microbiome composition in lean patients with NASH is associated with liver damage independent of caloric intake: A prospective pilot study. Nutr Metab Cardiovasc Dis 28: 369-84.

6    Chen, F.; Esmaili, S.; Rogers, G.B.; Bugianesi, E.; Petta, S.; Marchesini, G.; Bayoumi, A.; Metwally, M.; Azardaryany, M.K.; Coulter, S.; et al. (2020) Lean NAFLD: A Distinct Entity Shaped by Differential Metabolic Adaptation. Hepatology 71: 1213-27.

7    Mora-Gonzalez, J.; Migueles, J.H.; Esteban-Cornejo, I.; Cadenas-Sanchez, C.; Pastor-Villaescusa, B.; Molina-Garcia, P.; Rodriguez-Ayllon, M.; Rico, M.C.; Gil, A.; Aguilera, C.M.; et al. (2019) Sedentarism, Physical Activity, Steps, and Neurotrophic Factors in Obese Children. Med Sci Sports Exerc 51: 2325-33.

8    Che, D.; Liu, Q.; Rasheed, K., Tao, X. (2011) Decision tree and ensemble learning algorithms with their applications in bioinformatics. Adv Exp Med Biol 696: 191-9.

9    Weber, R.M., Fajen, B.R. (2015) Decision-tree analysis of control strategies. Psychon Bull Rev 22: 653-72.

10  Hashimoto, K. (2016) Ethnic differences in the serum levels of proBDNF, a precursor of brain-derived neurotrophic factor (BDNF), in mood disorders. Eur Arch Psychiatry Clin Neurosci 266: 285-7.

11   Wu, C.; Lu, J.; Lu, S.; Huang, M., Xu, Y. (2020) Increased ratio of mature BDNF to precursor-BDNF in patients with major depressive disorder with severe anhedonia. J Psychiatr Res 126: 92-7.

12  Siuda, J.; Patalong-Ogiewa, M.; Zmuda, W.; Targosz-Gajniak, M.; Niewiadomska, E.; Matuszek, I.; Jedrzejowska-Szypulka, H.; Lewin-Kowalik, J., Rudzinska-Bar, M. (2017) Cognitive impairment and BDNF serum levels. Neurol Neurochir Pol 51: 24-32.

13  Roy, N.; Barry, R.J.; Fernandez, F.E.; Lim, C.K.; Al-Dabbas, M.A.; Karamacoska, D.; Broyd, S.J.; Solowij, N.; Chiu, C.L., Steiner, G.Z. (2020) Electrophysiological correlates of the brain-derived neurotrophic factor (BDNF) Val66Met polymorphism. Sci Rep 10: 17915.

14  Fischer, D.L.; Auinger, P.; Goudreau, J.L.; Cole-Strauss, A.; Kieburtz, K.; Elm, J.J.; Hacker, M.L.; Charles, P.D.; Lipton, J.W.; Pickut, B.A.; et al. (2020) BDNF rs6265 Variant Alters Outcomes with Levodopa in Early-Stage Parkinson's Disease. Neurotherapeutics 17: 1785-95.

Reviewer 3 Report

The authors reported that BDNF was correlated with the reduction of activity in NAFLD patients and was the clinical important factor for the prevention and treatment of NAFLD.  The number of the subject is small, but this manuscript is important in the practice of NAFLD patients. Although the findings are pretty interesting, there is several concerns which need to be addressed.

Major revision

1.Are there the factors affected BNDF. How is the relationship with liver         fibrosis markers (i.e, liver stiffness, M2BPGi)?

2.The authors addressed that BDNF may be an important factor for the           prevention and treatment of NAFLD. The clinical usefulness of BDNF             should be discussed a little more.  Does it means that the NAFLD                 patients with BDNF level ≥ 19.1 ng/ml and HbA1c ≥ 7.1% need to                 treat?  

Minor revision

  1. The heading in 2.3(Patient characteristics) is wrong. It should be corrected.
  2. IRB approval no. of this study should be reported.
  3. The heading in 4.4 and 4.5 are the same title. It should be corrected.
  4. Does the following paragraphs need to devided? 4.7, 4.9 and 4.10 paragraph. If possible, these paragraph can be one paragraph.
  5. How about can 4.11~4.11.3 also be one paragraph?

Author Response

To REVIEWER 3

Thank you very much for your letter regarding our manuscript (life-1107884). We appreciate your comments, which have helped us to improve our manuscript. In line with your comments, please find below our response.

Major revision

Comment 1: Are there the factors affected BNDF. How is the relationship with liver fibrosis markers (i.e, liver stiffness, M2BPGi)?

Response: We appreciate your valuable comment. Following your suggestion, we have investigated the relationship between BNDF and liver fibrosis markers. There was a significant negative correlation of BDNF with liver stiffness and the FIB-4 index. However, these R2 values were approximately 0.10, which is generally considered negligible (Supplementary Figure 1 A and B).[1] In addition, no significant correlation was seen between BDNF and M2BPGi (Supplementary Figure 1C). These findings indicated that hepatic fibrosis may not be a major regulator for serum BDNF level in patients with NAFLD. We have added these data in the Discussion section (lines 191-193). Again, we appreciate your comments, which have helped us to improve our manuscript.

Supplementary Figure 1

Comment 2: The authors addressed that BDNF may be an important factor for the prevention and treatment of NAFLD. The clinical usefulness of BDNF should be discussed a little more.  Does it means that the NAFLD patients with BDNF level ≥ 19.1 ng/ml and HbA1c ≥ 7.1% need to treat? 

Response: We appreciate your valuable comment. As you indicated, “BDNF level ≥ 19.1 ng/mL and HbA1c ≥ 7.1%” is a profile for the high prevalence of reduced activity in patients with NAFLD. Since a reduction of activity is a risk factor for disease progression and mortality of patients with NAFLD,[2, 3] patients with such a profile may be an important target for the prevention and treatment of NAFLD. We have added this discussion in the revised manuscript (line 215-219).

Minor revision

Comment 1: The heading in 2.3(Patient characteristics) is wrong. It should be corrected.

Response: We appreciate your careful proofreading. We have corrected it as “Independent factors associated with the reduction of activity in patients with NAFLD” (line 115).

Comment 2: IRB approval no. of this study should be reported.

Response: We apologize that we did not provide IRB approval no. In the revised manuscript, we added the approval number (#20226) (line 238).

Comment 3: The heading in 4.4 and 4.5 are the same title. It should be corrected.

Response: We appreciate your careful proofreading. We have corrected the heading in 4.5. as “Disease-specific patient-reported outcome measure for patients with NAFLD” (line 265).

Comment 4: Does the following paragraphs need to devided? 4.7, 4.9 and 4.10 paragraph. If possible, these paragraph can be one paragraph.

Response: We agree with your comment. We combined 4.7, 4.9, and 4.10 paragraphs as one paragraph.

Comment 5: How about can 4.11~4.11.3 also be one paragraph?

Response: We appreciate your comment. We combined 4.11.1, 4.11.2, and 4.11.3 paragraphs as one paragraph following your suggestion.

References

  1. Heiman, G. Basic Statistics for the Behavioral Sciences. Correlation Coefficients: Wadsworth Cengage Learning 2011; 135-6.
  2. Romero-Gomez, M.; Zelber-Sagi, S., Trenell, M. Treatment of NAFLD with diet, physical activity and exercise. J Hepatol. 2017; 67: 829-46.
  3. Golabi, P.; Gerber, L.; Paik, J.M.; Deshpande, R.; de Avila, L., Younossi, Z.M. Contribution of sarcopenia and physical inactivity to mortality in people with non-alcoholic fatty liver disease. JHEP Rep. 2020; 2: 100171.

Round 2

Reviewer 1 Report

All comments were addressed correctly.

Authors can include a few lines in the discussion about why they did not chose IL1 or IL6 or any other inflammatory markers despite their higher levels in circulation. Authors can also elaborately discuss the importance of the statistical methods used here to identify and predict the genes/proteins of interest.

Author Response

To REVIEWER 1

Thank you very much for your letter regarding our manuscript (life-1107884). We appreciate your comments, which have helped us to improve our manuscript. In line with your comments, please find below our response.

Comment 1: Authors can include a few lines in the discussion about why they did not chose IL1 or IL6 or any other inflammatory markers despite their higher levels in circulation.

Response: We appreciate your comment. In this study, we have evaluated inflammatory and anti-inflammatory cytokines including IL-6, IL-1, IL-10, and IL-15. However, none of these inflammatory cytokines was identified as an independent factor for reduced activity. We have added these descriptions in the revised manuscript (lines 162-165).

Comment 2: Authors can also elaborately discuss the importance of the statistical methods used here to identify and predict the genes/proteins of interest.

Response: We appreciate your comment. As you pointed out, we did not provide information on the importance of data-mining analysis. In the revised manuscript, we have provided the following information (lines 72-84). Again, we appreciate your comments, which have helped us to improve our manuscript.

Physical function is regulated by diverse factors. A data-mining analysis is an artificial intelligence approach to reveal factors, even if no a priori hypothesis has been imposed.[1] This approach allows us to discover hidden factors that cannot be identified by logistic regression analysis.[2] Decision tree analysis is a data-mining technique and identifies factors with priorities. Thus, decision tree analysis is useful to discover the most impacting factor associated with physical function. Random forest analysis is also a data-mining technique used to identify factors that distinguish between case and control groups by random sampling. A feature of random forest analysis is a high level of predictive accuracy with an estimation of the relative importance for each factor [3]. Recently, these data-mining techniques have been used to identify individuals at an increased risk of developing NAFLD [4] and prognostic algorithms for patients with NAFLD-related hepatocellular carcinoma. [5] However, these unique statistical techniques have never been used to investigate the factor associated with physical function.

References

  1. Bellazzi, R., Zupan, B. Predictive data mining in clinical medicine: current issues and guidelines. Int J Med Inform. 2008; 77: 81-97.
  2. Yamada, S.; Kawaguchi, A.; Kawaguchi, T.; Fukushima, N.; Kuromatsu, R.; Sumie, S.; Takata, A.; Nakano, M.; Satani, M.; Tonan, T.; et al. Serum albumin level is a notable profiling factor for non-B, non-C hepatitis virus-related hepatocellular carcinoma: A data-mining analysis. Hepatol Res. 2014; 44: 837-45.
  3. Touw, W.G.; Bayjanov, J.R.; Overmars, L.; Backus, L.; Boekhorst, J.; Wels, M., van Hijum, S.A. Data mining in the Life Sciences with Random Forest: a walk in the park or lost in the jungle? Brief Bioinform. 2013; 14: 315-26.
  4. Perveen, S.; Shahbaz, M.; Keshavjee, K., Guergachi, A. A Systematic Machine Learning Based Approach for the Diagnosis of Non-Alcoholic Fatty Liver Disease Risk and Progression. Sci Rep. 2018; 8: 2112.
  5. Kawaguchi, T.; Tokushige, K.; Hyogo, H.; Aikata, H.; Nakajima, T.; Ono, M.; Kawanaka, M.; Sawada, K.; Imajo, K.; Honda, K.; et al. A Data Mining-based Prognostic Algorithm for NAFLD-related Hepatoma Patients: A Nationwide Study by the Japan Study Group of NAFLD. Sci Rep. 2018; 8: 10434.

Reviewer 2 Report

I am sorry to say that I cannot see differences between the original and the V2 of the manuscript, despite what stated in the Rebuttal letter. Based on that my comment and opinion are unchanged. Moreover, the fact that you are not able to enroll more subjects is not a sufficient reason for accepting a work that need an increase in the power of the analysis.

Author Response

To REVIEWER 2

Thank you very much for your letter regarding our manuscript (life-1107884). We appreciate your comments, which have helped us to improve our manuscript. In line with your comments, please find below our response.

Comment 1: I am sorry to say that I cannot see differences between the original and the V2 of the manuscript, despite what stated in the Rebuttal letter. Based on that my comment and opinion are unchanged. Moreover, the fact that you are not able to enroll more subjects is not a sufficient reason for accepting a work that need an increase in the power of the analysis.

Response: We apologize that we uploaded the wrong file in the previous revision. We have revised the following 7 points in the revised manuscripts according to your previous comments. As you pointed out, we agree that there is a potential bias because of the small sample size, lacking a control group, experimental plan, and sex difference in this study. Therefore, we have described these issues and revised them as much as possible in the revised manuscript. We would appreciate it if you could understand the following explanations.

  • Small sample size

Response: As you pointed out, we agree that a small sample size, stratification for two groups (normal activity or reduced activity), and lacking a control group are potential biases in the analysis. However, this study is designed as a nested case-control study of the NAFLD cohort and we are not able to increase the number of enrolled subjects at present. It is also difficult to enroll healthy control in this study at present. Therefore, we added these issues as limitations of this study (lines 220-221).

  • A potential bias in the experimental plan

Response: As you pointed out, this study has a potential bias in the experimental plan. Therefore, we investigated factors associated with the reduced activity by exploratory data-mining analyses using various factors including age, sex, BMI, HbA1c, liver stiffness, and various cytokines. As a result, the association between reduced activity and serum BDNF level was confirmed by two types of data-mining analyses such as decision-tree and random forest analyses. Thus, we tried to eliminate possible bias by using various factors and multiple exploratory data-mining analyses. We have added the above descriptions in the revised manuscript (lines 162-164, line 168).

  • A high male ratio in the reduced activity group

Response: As you pointed out, the male-to-female ratio was significantly higher in the reduced activity group than in the normal activity group. The unsuccessful result in Figure 2 may be related to the low statistical power as you suggested. However, Cohen et al. recently investigated gender differences in physical activity of general subjects and demonstrated that physical activity level is significantly higher in men than in women [1]. Moreover, in decision-tree analysis, sex was not identified as a classifier with the reduced activity in patients with NAFLD (Figure 3). Furthermore, in random forest analysis, sex (male) was the fifth-ranked variable for distinguishing between the reduced and normal activity groups in patients with NAFLD (Figure 4). Thus, in patients with NAFLD, the impact of sex on physical activity may be different from the general population and may have less impact on physical activity level. We have added the above discussion in the revised manuscript (lines 152-161).

  • Association between BDNF level and the intensity of physical activity

Response: We apologize for the unclear statement. BDNF level is linked to the intensity of physical activity in patients with who are overweight/obese, but not obesity itself [2]. In the revised manuscript, we clearly stated this issue (lines 68-70).

  • The cut-off value was determined by the decision-tree analysis

Response: We apologize for the insufficient description of the cut-off value for BDNF and HbA1c. The cut-off value was determined by the decision-tree analysis. The decision-tree analysis is a data-mining analysis, which gives the best split of valuable [3, 4]. We have added this point in the revised manuscript (lines 316-317).

  • Revision of the first sentence of the discussion

Response: We appreciate your proper suggestion. As you pointed out, activity was assessed by the activity domain of CLDQ-NAFLD/NASH in this study. According to your suggestion, we have revised the sentence as follows: “in this study, reduction of the activity domain of CLDQ-NAFLD/NASH was seen in 43,8% of patients with NAFLD.” (lines 146-147).

  • Enlarge the section of the discussion

Response: We appreciate your comment. As you suggested, our explanation of the controversial result is not based on direct evidence and is an overstatement. Following your suggestion, we have revised this paragraph by replacing alternative explanations as follows: “We could not examine the causal relationship between BNDF level and reduction of activity in patients with NAFLD in this study. Although the biological significance of BNDF in the reduction of activity remains unclear, BDNF has been reported to be modulated by various factors including race [5], depressive symptoms [6], cognitive function [7], and genetic polymorphism [8, 9]. These factors might be involved in the negative association between serum BDNF levels and activity in this study.” (lines 203-208). Again, we appreciate your constructive comments, which have helped us to improve our manuscript.

References

  1. Cohen, D.A.; Williamson, S., Han, B. Gender Differences in Physical Activity Associated with Urban Neighborhood Parks: Findings from the National Study of Neighborhood Parks. Womens Health Issues. 2020:
  2. Mora-Gonzalez, J.; Migueles, J.H.; Esteban-Cornejo, I.; Cadenas-Sanchez, C.; Pastor-Villaescusa, B.; Molina-Garcia, P.; Rodriguez-Ayllon, M.; Rico, M.C.; Gil, A.; Aguilera, C.M.; et al. Sedentarism, Physical Activity, Steps, and Neurotrophic Factors in Obese Children. Med Sci Sports Exerc. 2019; 51: 2325-33.
  3. Che, D.; Liu, Q.; Rasheed, K., Tao, X. Decision tree and ensemble learning algorithms with their applications in bioinformatics. Adv Exp Med Biol. 2011; 696: 191-9.
  4. Weber, R.M., Fajen, B.R. Decision-tree analysis of control strategies. Psychon Bull Rev. 2015; 22: 653-72.
  5. Hashimoto, K. Ethnic differences in the serum levels of proBDNF, a precursor of brain-derived neurotrophic factor (BDNF), in mood disorders. Eur Arch Psychiatry Clin Neurosci. 2016; 266: 285-7.
  6. Wu, C.; Lu, J.; Lu, S.; Huang, M., Xu, Y. Increased ratio of mature BDNF to precursor-BDNF in patients with major depressive disorder with severe anhedonia. J Psychiatr Res. 2020; 126: 92-7.
  7. Siuda, J.; Patalong-Ogiewa, M.; Zmuda, W.; Targosz-Gajniak, M.; Niewiadomska, E.; Matuszek, I.; Jedrzejowska-Szypulka, H.; Lewin-Kowalik, J., Rudzinska-Bar, M. Cognitive impairment and BDNF serum levels. Neurol Neurochir Pol. 2017; 51: 24-32.
  8. Roy, N.; Barry, R.J.; Fernandez, F.E.; Lim, C.K.; Al-Dabbas, M.A.; Karamacoska, D.; Broyd, S.J.; Solowij, N.; Chiu, C.L., Steiner, G.Z. Electrophysiological correlates of the brain-derived neurotrophic factor (BDNF) Val66Met polymorphism. Sci Rep. 2020; 10: 17915.
  9. Fischer, D.L.; Auinger, P.; Goudreau, J.L.; Cole-Strauss, A.; Kieburtz, K.; Elm, J.J.; Hacker, M.L.; Charles, P.D.; Lipton, J.W.; Pickut, B.A.; et al. BDNF rs6265 Variant Alters Outcomes with Levodopa in Early-Stage Parkinson's Disease. Neurotherapeutics. 2020; 17: 1785-95.

Reviewer 3 Report

The author sincerely responded to the reviewers' comments. The manuscript is worth publishing.